# Effects of Long-Term Gentle Handling on Behavioral Responses, Production Performance, and Meat Quality of Pigs

**DOI:** 10.3390/ani10020330

**Published:** 2020-02-19

**Authors:** Chao Wang, Yongjie Chen, Yanju Bi, Peng Zhao, Hanqing Sun, Jianhong Li, Honggui Liu, Runxiang Zhang, Xiang Li, Jun Bao

**Affiliations:** 1College of Animal Science and Technology, Northeast Agricultural University, Changjiang Road No. 600, Harbin 150030, Heilongjiang, China; wqclly@163.com (C.W.); chenyongji12@163.com (Y.C.); yanju_bi@163.com (Y.B.); 15146009893@163.com (P.Z.); sunhq08@126.com (H.S.); liuhonggui1312@163.com (H.L.); zhangrunxiang@neau.edu.cn (R.Z.); 2College of Life Science, Northeast Agricultural University, Changjiang Road No. 600, Harbin 150030, Heilongjiang, China; jhlineau@126.com; 3Key Laboratory of Swine Facilities, Ministry of Agriculture, Northeast Agricultural University, Changjiang Road No. 600, Harbin 150030, Heilongjiang, China

**Keywords:** gentle handling, behavioral response, production performance, meat quality, pig

## Abstract

**Simple Summary:**

As an important part of modern livestock production, human–animal relationships influence the welfare of farm animals. Previous studies have shown that short-term gentle handling can reduce pigs’ anxiety, and improve production performance and possibly meat quality. However, the effectiveness of long-term gentle handling is still unknown. Therefore, this study aimed to investigate the impact of gentle handling of growing pigs over a relatively long period of time on their behavior, production performance, and meat quality after slaughter. Our results show that gentle handling increased intimacy between the handler and handled pigs, whereas long-term gentle handling had little effect on pig production performance, or on carcass and meat quality. On the other hand, long-term gentle handling had positive effects on production performance, and reduced pigs’ anxiety and increased their willingness to approach the handler. Hence, this study provides insights into the positive effects of long-term human–animal interactions on pig production.

**Abstract:**

In order to investigate the effect of gentle handling on the behavior, performance, and meat quality of pigs from weaning to slaughter, 144 6-week-old weaned hybrid Min piglets (a native breed) were selected and divided into a handling group (HG: 9 pens × 8 pigs) and a control group (CG: 9 pens × 8 pigs). After 6 weeks of handling, we observed and then evaluated the pigs’ responses to a handler with behavioral scores. Moreover, we measured heart rate and production performance. Three pigs were randomly selected from each of the 18 pens and divided into a handling group (HG: *n* = 27) and a control group (CG: *n* = 27), and the HG pigs were given gentle handling until slaughter. Subsequently, we evaluated meat quality and the production performance of six pigs from each group. The results show that AA test (approaching–avoidance test) scores in HG pigs, the number of contacts with the handler and absence of contact with the handler, were significantly higher than in the CG pigs (*p* < 0.01). The occurrences of avoidance and looking at the handler were lower in the HG than in the CG group (*p* < 0.01); however, heart rate was not found to be significantly different between the two groups (*p* = 0.63). No significant difference was found in the average daily gain, average daily feed intake, and feed conversion ratio between the two groups during the two periods (*p* > 0.05). The b* value was determined 45 min after slaughter, and it was significantly lower in the HG than that in the CG group (*p* = 0.002). Furthermore, 2 h after slaughter, the L value of the HG group was significantly higher than that of the CG group (*p* = 0.047), but no difference was observed in carcass quality or other meat quality indicators between the two groups (*p* > 0.05). The results indicate that gentle handling could reduce pigs’ anxiety and increase their willingness to approach the handler, increasing the intimacy of the pigs and handlers. However, long-term gentle handling had little effect on pig performance, carcass quality, and meat quality.

## 1. Introduction

Human–animal relationships are an important part of management in modern husbandry production, and an important component of farm animal welfare [1]. Poor treatment of farm animals increases the anxiety of animals and causes difficulties in their handling and management. This worsens the human–animal relationship and has a negative impact on production performance and welfare [2,3]. The manner of pigs’ handling can determine the animals’ attitude towards handlers, as gently handled pigs are less afraid of people than badly treated ones. In addition, mixed poor and good treatment has been shown to have an identical effect to poorly handled treatment [4]. A series of studies [4,5,6,7] have confirmed that pigs with a high degree of fear of humans have restricted growth and production performance due to a chronic stress response. One of the main indicators of a good human–animal relationship is a low stress response of the animals towards people, and increased willingness to approach and make contact with people [3]. Gentle or positive handling can be a potential way to reduce pigs’ anxiety caused by interactions with humans. Several studies have reported that after 5–10 weeks of gentle handling, the time required for pigs to approach people was shorter, while the frequency of contact and communication with people increased [4,5,6,7,8]. Furthermore, touching and scratching the animals can improve their relationship with handlers, which consequently alters the heart rate of pigs and increases the intimacy between pigs and humans [9]. Gentle handling can improve the daily feed intake [10], feed conversion rate [11], and growth efficiency [4] of weaned piglets during a certain period. Likewise, gentle handling can affect pigs’ stress before slaughtering, and possibly improve the quality of the meat [12,13].

Thus, gentle handling can reduce the fear of humans, thereby improving animal welfare and production performance, and possibly meat quality [14]. However, recent studies on human–pig interactions have been focused on short-term impacts during the growing or final period. Therefore, this study aimed to investigate the impact of gentle handling on the behavior of growing pigs over a relatively longer time, as well as its impact on the production performance and meat quality after slaughter.

## 2. Materials and Methods

### 2.1. Animals and Management

One hundred and forty-four 6-week-old crossbred piglets (Large White × Large White × Duroc × Min) weaned at 5 weeks of age were selected from 18 litters, with a mean age of 47.7 ± 3.7 days, and allocated into 18 pens with 4 male and 4 female piglets in each pen with a mean body weight (BW) of 10.9 ± 0.68 kg. The 144 piglets were divided into two groups (9 pens/group), namely a gentle handling group (HG) and a control group (CG). All piglets wore ear tags for individual identification. The experiment was set for two periods. Phase I: Approximately 6 weeks from 8 to 14 weeks of age. Phase II: A period from 15 weeks of age until slaughter at weight about 110 kg. 

In Phase I, all piglets were housed in the same shed with a pen size of 4.0 m (length) × 2.4 m (width) × 1.2 m (height) with a concrete floor. Heat supplement was supplied with a heating bulb (2000 W) over the pens at 0.5 m height above the floor. Visual contact between the two groups was prevented with a 1.2 m high fence in the middle passage, built out of a 5 mm-thick high-density board. The piglets had ad libitum access to feed and water. The feed formula was 11.87 MJ/kg Digestible Energy (DE), 15.5% Crude Protein (CP), 4.0% Crude Fibre (CF) and 1.3% Lysine (Lys). The pens were cleaned twice a day, at 7.00 a.m. and 4.30 p.m. The range of temperature and humidity was 15–21 ℃ and 55–70%, respectively.

In Phase II, three piglets were randomly selected from each pen, and a total of 54 pigs (3 pigs/pen × 18 pens) were selected. These 54 piglets were randomly divided into two groups: HG (3 pens × 9 pigs from group HG in Phase I) and CG (3 pens × 9 pigs from group CG in Phase I). The pen sizes of both groups were 4.0 m × 4.0 m × 1.1 m. Each pen was equipped with an electronic feeding station (Osborne, USA). The pigs were fed with a formula of 13.02 MJ/kg DE, 17.0% CP, 3.40% CF, and 1.0% Lys until the end of the fattening period. They had ad libitum access to water. Excrements were cleaned twice a day at 5.00 a.m. and 1.00 p.m., and the pen floor was disinfected once weekly. The house temperature was controlled with natural ventilation. During the experimental period, the range of temperature and humidity was 12–25 °C and 50%–75%, respectively.

### 2.2. Experiment Design

#### 2.2.1. Gentle Handling Treatment

A male handler dressed in a green camouflage suit and shoes was responsible for each handling treatment in the HG group during the handling period (from 8 to 13 weeks of age and from 15 weeks of age until slaughter). The gentle handling treatment was performed in the HG group from 8.00 a.m. to 10.00 a.m. and from 2.00 p.m. to 4.00 p.m. for 5 days of a week in Phase I and II, according to the method presented by Tallet et al. [9]. In the treatment group, a handler remained standing still for 2 min after entering the HG pen, and then knelt and gently touched the pigs; the contact lasted for more than 1 min with each pig. During the touching period, the handler scratched the pigs’ head, or neck or back with their fingers and spoke to them. The handler spent about 10 min in each HG pen. During the whole experiment, no one could enter the CG pens except for a worker who cleaned the pen prior to the handling. During the cleaning process, the worker was forbidden from having any visual or bodily contacts with the pigs. 

#### 2.2.2. Approaching–Avoidance Test

The approaching–avoidance (AA) test was carried out at 14 weeks of age in a testing pen after the handling treatment, and the test pen was in the same house. Before the formal AA test, a training program for acclimation to the test pen was performed. This was done by bringing all testing pigs into three new pens which were identical to the testing pen. During each training, the pig stayed in the testing pen for 30 min and then was taken back to its home pen. Every subsequent pig followed an identical procedure, one after the other. The pigs were trained simultaneously for 30 min daily to wear heart rate (HR) belts for continuous HR recording, before they entered the testing pens. This training program lasted for two consecutive days. After every training session, the belts were removed immediately, and they were brought back to their home pens.

Two days of training were followed by the AA test. To observe the pigs’ reactions to the handler, a pig was brought into one of the three test pens and the willingness of the animal to approach a handler in a novel environment was considered as an indicator of a good human–animal relationship [15]. The testing procedure was performed as previously described by Forde et al. [16]. After the pig spent 10 min in the testing pen, the handler, wearing the same clothes as usual, entered the testing pen and stood still at the door for 1 min. If the pig hesitates to approach the handler within 1 min, the handler would slowly move forward, outstretch his hand towards the pig, and attempt to touch the pig. If the pig turns its head aside or retreats, then the test is terminated, and a score of avoidance is given. The scores for difference levels of avoidance and approach reactions were given in Table 1. In this test, 36 pigs were participated, and each was tested only once.

After the completion of the AA test, the handler returned to the test pen and the piglets’ behavioral responses to the hander were recorded for 5 min for all 36 pigs. The piglets’ behavioral parameters are listed in Table 2. Their behavioral responses were recorded with a video camera (DS-7816N-E2, Hikvision Digital Technology Co., Ltd, Hangzhou, China). For behavioral sampling, scan animal sampling and instantaneous recording methods were adopted, and the behavioral responses listed were sampled at 3 s intervals.

#### 2.2.3. Heart Rate (HR) Measurement

Heart rate measurement was carried out, along with the AA test, with a Bluetooth heart rate belt (297 mm × 32 mm × 12 mm, 41.4 g, SMART BELT, Decathlon Group, Shanghai, China). The HR belts were assembled before the pigs entered the testing pens. To ensure that the belt sensor worked properly, the hairs under the left forelimb around the heart area were shaved and the belt was placed around the piglets’ chests, with the sensor connecting with the shaved skin. The heart rate sensor was connected with ALA coach+ software (ver.2.2.1, Android, ALATECH Technology Limited, Taiwan, China) via Bluetooth. HR recording started when the handler entered the testing pen and ended after the handler went out of the testing pen. The average bpm (beats per min) recorded on ALA coach+ software represents the heart rate data for the AA test, during which the handler remained in the testing pen. The belts were removed after completion of the test. The mean HR was calculated using the total recorded data over the testing period. 

#### 2.2.4. Production Performance Measurement

The average daily weight gain (ADWG) and average daily feed intake (ADFI) during phase I were calculated based on the daily feed weight, the remaining feed in the feeding trough, and the weight of piglets on the 7th day of week 5 and the 7th day of week 13. The feed conversion ratio (FCR) was calculated according to the daily weight gain and daily feed intake. ADWG, ADFI, and FCR were measured with Feed Intake Recording Equipment (FIRE2.2.0.8, Osborne Industrial Group, Shanghai, China) in phase II. 

#### 2.2.5. Carcass Quality

Two pigs (12 pigs, HG: *n* = 6, 103.07 ± 6.60 kg; CG: *n* = 6, 101.17 ± 2.26 kg) from six pens in phase II were selected and slaughtered. Prior to slaughter, the pigs fasted for 24 h. The pigs were stunned with 85 V for 15 s and the blood was collected afterward. The heads, hooves, tails, and viscera were removed after slaughter. The skin, bone, lean meat, and fat were separated from the left carcass and weighed. The slaughter rate, average backfat thickness, lean meat rate, eye muscle area, and fat rate were determined according to previously reported methods [17]. 

#### 2.2.6. Meat Quality

The longissimus dorsi muscle at the thoracolumbar junction of the left carcass was removed immediately, and used to measure the meat pH value (45 min, 2 h, 24 h, 48 h) [12] and meat color (L, a*, b*, 45 min, 2 h, 24 h, 48 h). The longest dorsi muscles from the 3rd to 6th lumbar vertebrae were taken from the left parts of the bodies and used to measure dripping loss, squeezing loss, and cooking loss [12]. 

### 2.3. Statistical Analysis

The original data were processed by Excel 2016 (Microsoft Corporation, Redmond, Washington, USA) and SPSS 25.0 (IBM Corporation, Armonk, NY, USA). The single-sample Kolmogorov–Smimov process was performed prior to statistical analysis to confirm the normal distribution of the behavioral data. A one-way ANOVA analysis (SPSS 25.0) was used to analyze the data. The results are expressed as the mean ± standard deviation, and *p* < 0.05 was considered significant. 

## 3. Results

### 3.1. Effects of Gentle Handling on Behavioral Responses of Pigs

The data in Table 3 represent the influence of gentle handling on pigs’ behavioral responses to the handler. The results indicate that the HG pigs had higher AA test scores than the CG pigs regarding the occurrence of contact, absence of the contact, decreased visual contact with the handler, and avoidance (*p* < 0.01), and a significantly lower incidence of looking and avoidance behaviors than the CG pigs (*p* < 0.01). HR was not significantly different between the HG and CG groups (*p* = 0.63).

### 3.2. Effect of Gentle Handling on Production Performance of Pigs

No significant difference was found in ADWG (5.12 ± 1.02 for HG, 2.33 ± 1.18 for CG, *p* = 0.86), ADFI (0.84 ± 0.26 for HG, 0.82 ± 0.24 for CG, *p* = 0.7), or FCR (2.16 ± 0.61 for HG, 1.98 ± 0.31 for CG, *p* = 0.2) between the HG and CG groups in phase I, as shown in Table 4, nor in ADWG (0.67 ± 0.11 for HG, 0.71 ± 0.13 for CG, *p* = 0.22), ADFI (2.29 ± 0.35 for HG, 2.36 ± 0.24 for CG, *p* = 0.55), or FCR (3.48 ± 0.39 for HG, 3.39 ± 0.60 for CG, *p* = 0.51) during phase II, as shown in Table 5.

### 3.3. Effects of Gentle Handling on Carcass and Meat Quality

No significant difference was found in slaughter rate (73.84 ± 2.96 vs. 71.92 ± 2.26, *p* = 0.24), backfat thickness (25.72 ± 4.07 vs. 22.09 ± 6.97, *p* = 0.30), lean meat rate (58.36 ± 4.60 vs. 60.98 ± 5.96, *p* = 0.42), eye muscle area (32.84 ± 7.40 vs. 34.05 ± 7.23, *p* = 0.78), or fat ratio (23.50 ± 5.82 vs. 20.11 ± 6.83, *p* = 0.38) between the HG and CG pigs, as shown in Table 6. 

Although the pH value was found to be higher in HG than CG pigs, the difference was not significant (45 min) (6.47 ± 0.26 vs. 6.42 ± 0.17, *p* = 0.52), pH (2 h) (6.20 ± 0.40 vs. 6.03 ± 0.32, *p* = 0.14), pH (24 h) ( 5.56 ± 0.06 vs. 5.54 ± 0.09, *p* = 0.29), pH (48 h) ( 5.53 ± 0.06 vs. 5.52 ± 0.07, *p* = 0.64), as shown in Table 7. 

Table 8 shows the results of gentle handling on meat color. No significant difference was found between the HG and CG pigs 45 min after slaughter (45.63 ± 1.43 vs. 45.60 ± 1.39 for L, *p* = 0.96; 3.78 ± 0.29 vs. 3.79 ± 0.75, *p* = 0.98); however, the b* value in the HG pigs was significantly lower than that in the CG group (1.22 ± 0.18 vs. 1.71 ± 0.23, *p* < 0.01). Furthermore, the L value was significantly lower in the HG than in the CG group 2 h after the slaughter (43.34 ± 3.10 vs. 47.06 ± 2.91, *p* < 0.05), while no difference was detected in the a* and * values (3.09 ± 0.97 vs. 3.69 ± 1.64, *p* = 0.46 for a*; 1.31 ± 0.45 vs. 2.20 ± 1.20, *p* = 0.12 for b*). At 24 h and 48 h, no significant differences were found in L, a*, or b* values (52.75 ± 2.15 vs. 53.27 ± 2.64, *p* = 0.71 for L (24 h); 6.25 ± 1.08 vs. 5.91 ± 1.37, *p* = 0.64 for a*(24 h); 2.61 ± 0.95 vs. 2.61 ± 0.73, *p* = 0.99 for b* (24 h); 53.39 ± 2.14 vs. 53.96 ± 2.93, *p* = 0.74 for L (48 h); 6.90 ± 0.88 vs. 6.51 ± 2.06, *p* = 0.68 for a* (48 h); 3.75 ± 0.63 vs. 3.46 ± 1.06, *p* = 0.58 for b* (48 h)). 

Similarly, no significant differences were found in squeezing loss (0.13 ± 0.10 vs. 0.09 ± 0.05, *p* = 0.35), dripping loss (0.06 ± 0.03 vs. 0.04 ± 0.02, *p* = 0.42), or cooking loss (0.36 ± 0.32 vs. 0.35 ± 0.03, *p* = 0.40) between the two groups, as shown in Table 9. 

## 4. Discussion

### 4.1. Effects of Gentle Handling on Pigs’ Behavior

After 6 weeks of gentle handling treatment with consistent frequency, the pigs in the HG group obtained a higher AA test score than the CG pigs, which is consistent with the results of Tallet [9], who found a 25-day treatment of gentle handling contributed to the establishment of a positive human–animal relationship and increased the pigs’ willingness to approach people. In addition, HG pigs made contact more often with the handler, including showing signs of a mother–child interaction—putting the front hooves on the handler’s legs—indicating the higher interest of these animals in their handler [18]. Conversely, the CG pigs made more visual contact with the handler but engaged in much less physical contact, and showed lower willingness to approach. This may be an indicator of a fearful response to the handler. Of note, the behavior of looking towards the handler was present and increased when a novel or aversive stimulus was initiated [19], indicating that the control pigs were unfamiliar with the handler and might have regarded the handler as a potential danger. These results indicate the increased intimacy between pigs and humans after gentle handling, which is consistent with the results of the abovementioned AA test score. 

The absence of contact with the handler was more often seen in HG pigs than that in the control group, and these pigs were more inclined to explore the pen. This observation is relevant to the hypothesis that pigs are more interested in exploring novel environments (the new pen) if they are not disturbed by the presence of the handler. Regardless of the similarity between the home pen and the testing pen, pigs naturally require time to become familiar with a new environment [9]. 

The results of this study indicate a slightly lower heart rate of the HG pigs than in the CG group, but the difference was not significant. This may suggest that although the CG pigs showed more fear or tension than the HG animals, this tension may not be enough to cause a significant variation in HR between the two groups. 

### 4.2. Effects of Gentle Handling on Pig Performance

For 6 weeks of gentle handling after weaning, the average daily feed intake and feed conversion ratio of the HG and CG pigs were different; however, the differences were not significant. These results are consistent with the findings of Hemsworth et al. [4], whose research was conducted on young pigs inconsistently handled by humans for 6 weeks, and with the results of Gonyou et al. [5], who studied regular interactions between humans and growing pigs for 10 weeks. Contrarily, Hemsworth et al. found that the 5-week treatment of aversively handled pigs individually or in groups decreased production performance [11]. These results indicate that gentle or positive handling, whether consistent or inconsistent, occasional or frequent, did not influence the production performance of pigs. However, the negative handling of pigs has an adverse effect on production performance. On the other hand, our study had a larger group size and smaller feeding density (8 pigs per pen, 1.2 m^2^ per pig) compared to Hemsworth et al. (5 pigs per pen, 1.7 m^2^ per pig) [11]. This may indicate that a smaller group sizes and more space give pigs more chances to approach people, which can reduce the fear and anxiety of pigs towards humans. During gentle handling throughout the final period, a difference in production performance between the HG and CG pigs was documented; however, it was not statistically significant, and this is consistent with the results of Gonyou et al. [5], Hemsworth et al. [4], Pearce et al. [20], Hemsworth et al. [11], and more. These studies show that after gentle handling for 10–13 weeks, pig production performance is not affected. Therefore, a pig’s production performance may be affected positively by gentle handling for a short time; however, these positive effects may disappear in the long term as the pigs adapt to the gentle handling treatment. 

### 4.3. Effects of Gentle Handling on Carcass and Meat Quality

Former studies have shown that increased feed intake and growth rate of finishing pigs could contribute to a gain in fat deposition, increased backfat thickness, reduced lean meat rate [17], and differences in the eye muscle area [21]. Although no significant differences were found in the feed intake and growth rate between the handling group and the control group, the results of this study show that the carcass quality reflected the abovementioned trend of the changes in carcass parameters.

There was no significant difference in meat pH values between the handling group and the control group at four time points after slaughter. The obtained results are consistent with the results of D ’souza et al., who studied on-farm and pre-slaughter handling of pigs [12]. Similar results were obtained by Hemsworth et al., who examined positive pre-slaughter handling for 5 weeks [13]. This may suggest that the time length of gentle handling does not affect the pH values at 45 min, 2 h, 24 h, and 48 h after slaughter. In addition, it is widely known that dripping loss, squeezing loss, and cooking loss reflect the water retention capacity of meat (the hydrodynamics). In this study, gentle handling had no significant effect on the hydrodynamics of muscles, which is consistent with the results of D ’souza et al. [12]. Previous studies have reported that the hydrodynamics of meat are affected by pH and the glycolysis process of meat after slaughter [22,23]. The effect of gentle handling on the glycolysis process of meat after the slaughter was not investigated in this study, and further research is required to understand the effect of gentle handling on the water loss of meat.

On the other hand, gentle handling had significant effects on the b* value and L value within 1 h after slaughter, while no significant effect on muscle color was found at the other tested time points. This is consistent with the research of Gonyou et al. [5] and D ’souza et al. [12], who studied the effect of pre-slaughter handling on meat color and other features. In their studies, after 5 weeks of positive handling, there was no significant difference in muscle color after slaughter; however, these studies were based on the L value only, or on a single time point. Therefore, gentle handling may have a small effect on muscle color.

### 4.4. Scientific Reasoning for the Improvement of Animal Welfare Legislation

Although there are no clear regulations around the human–animal relationship or interactions in the animal welfare laws of the EU or USA, regulations specifying that stockmen on pig farms should have a basic biological knowledge of pigs, and have certain management skills and basic veterinary skills during pig production and transportation, exist in EU laws (Council Regulation 1/2005, 2008/120/EC et al.; American Animal Welfare Act and the Welfare of Farmed Animals (England) Regulations 2007). In this study, additional gentle handling on the basic of regular management in pig production had little significant effect on production performance or meat quality, which means that gentle or positive handling may not influence production or pork quality of pigs. However, the effects of long-term gentle handling on pigs’ behavior and on several indexes of meat quality were obvious. In addition, studies on the effects of pre-slaughter negative handling have found evident negative effects on carcasses [24,25,26]. Those studies were different from ours in terms of handling type (negative vs. gentle) and time duration (40 days or loading before slaughter vs. from weaning to slaughter), which might be the reason for the results of those studies and our study being different. Therefore, long-term positive or gentle handling (additional caring, gentle tactile contacts, or even simple presence around the pigs) should be provided, or at least negative handling (abuse, electric shock, and so forth) should be prevented in pig production. Thus, our study might provide a scientific basis for the legislation of the human–animal relationship or interactions, and support the improvement of animal welfare laws.

## 5. Conclusions

After 6 weeks of gentle handling, pigs showed less fearfulness and more willingness to approach humans, as well as increased intimacy with the handler. However, long-term gentle handling (from weaning to slaughter) had little effect on pig production performance, carcass quality, and meat quality. Even so, it is necessary that additional positive or gentle handling should be provided in regular livestock management in order to reduce fear and improve animal welfare in modern pig production.

## Figures and Tables

**Table 1 animals-10-00330-t001:** Scoring standard for avoidance and approach [9,16].

The Behavioral Response of Piglets After the Handler Entered the Testing Pen	Scoring Criteria	Score
Avoidance	The handler reached out and approached the pig	1
	The handler stopped about 0.5 m between the handler and the pig	2
	The handler tried to touch pig	3
	The handler touched pig	4
Approach	Pig was in a 0.5 m range of the handler	5
	The pig came into contact with the handler	6

**Table 2 animals-10-00330-t002:** Parameters of behavioral responses of a pig to the handler [9,16].

Behavior	Definition
Look at the handler (looking)	Head pointing to the handler
Contact with the handler (contact)	Pigs touch or sniff handler; piglets interact with the handler by placing their forelegs on the handler
Avoiding handler (avoidance)	Move away from the handler, or turn head in the opposite direction from the handler; legs on the wall to try to escape the test pen
No contact with subjects (no contact)	Does not interact with the handler or explore the test pen

**Table 3 animals-10-00330-t003:** Effect of gentle handling on the behavioral responses and heart rate of pigs (mean ± standard deviation) (*p* < 0.05).

Behavior Classification	HG	CG	*p*-Value
AA test score	5. 12 ± 1.02	2.33 ± 1.18	<0.01
Looking at handler (times)	7.78 ± 4.87	17.33 ± 12.41	<0.01
Contact (times)	24.11 ± 21.25	4.44 ± 2.97	<0.01
Avoidance (times)	11.83 ± 15.87	38.94 ± 11.95	<0.01
No contact (times)	56.17 ± 21.90	36.00 ± 12.26	<0.01
Heart rate (BPM)	136.28 ± 5.45	137.78 ± 3.04	0.63

HG: gentle handling group; CG: control group.

**Table 4 animals-10-00330-t004:** Effect of gentle handling on production performance of pigs during phase I (mean ± standard deviation) (*p* < 0.05). ADWG: average daily weight gain; ADFI: average daily feed intake; FCR: feed conversion ratio.

Production Performance P phase I	HG	CG	*p*-Value
ADWG	0.43 ± 0.16	0.43 ± 0.14	0.86
ADFI	0.84 ± 0.26	0.82 ± 0.24	0.7
FCR	2.16 ± 0.61	1.98 ± 0.31	0.2

**Table 5 animals-10-00330-t005:** Effect of gentle handling on production performance of pigs during phase II (mean ± standard deviation) (*p* < 0.05).

Production Performance during phase II	HG	CG	*p*-Value
ADWG	0.67 ± 0.11	0.71 ± 0.13	0.22
ADFI	2.29 ± 0.35	2.36 ± 0.243	0.55
FCR	3.48 ± 0.39	3.39 ± 0.60	0.51

**Table 6 animals-10-00330-t006:** Effect of gentle handling on carcass quality of finishing pigs (mean ± standard deviation) (*p* < 0.05).

Carcass Quality	HG	CG	*p*-Value
Slaughter rate %	73.84 ± 2.96	71.92 ± 2.26	0.24
Backfat thickness mm	25.72 ± 4.07	22.09 ± 6.97	0.3
Lean meat rate %	58.36 ± 4.60	60.98 ± 5.96	0.42
Fat rate %	23.50 ± 5.82	20.11 ± 6.83	0.38
Eye muscle area cm²	32.84 ± 7.40	34.05 ± 7.23	0.78

**Table 7 animals-10-00330-t007:** Effect of gentle handling on meat pH of finishing pigs (mean ± standard deviation) (*p* < 0.05).

Meat pH	HG	CG	*p*-Value
pH(45 min)	6.47 ± 0.26	6.42 ± 0.17	0.52
pH(2 h)	6.20 ± 0.40	6.03 ± 0.32	0.14
pH(24 h)	5.56 ± 0.06	5.54 ± 0.09	0.29
pH(48 h)	5.53 ± 0.06	5.52 ± 0.07	0.64

**Table 8 animals-10-00330-t008:** Effect of gentle handling on meat color of finishing pigs (mean ± standard deviation) (*p* < 0.05).

Meat Color		HG	CG	*p*-Value
45 min	L	45.63 ± 1.43	45.60 ± 1.39	0.96
	a*	3.78 ± 0.29	3.79 ± 0.75	0.98
	b*	1.22 ± 0.18	1.71 ± 0.23	<0.01
2 h	L	43.34 ± 3.10	47.06 ± 2.91	<0.05
	a*	3.09 ± 0.97	3.69 ± 1.64	0.46
	b*	1.31 ± 0.45	2.20 ± 1.20	0.12
24 h	L	52.75 ± 2.15	53.27 ± 2.64	0.71
	a*	6.25 ± 1.08	5.91 ± 1.37	0.64
	b*	2.61 ± 0.95	2.61 ± 0.73	0.99
48 h	L	53.39 ± 2.14	53.96 ± 2.93	0.74
	a*	6.90 ± 0.88	6.51 ± 2.06	0.68
	b*	3.75 ± 0.63	3.46 ± 1.06	0.58

**Table 9 animals-10-00330-t009:** Effect of gentle handling on water loss of meat (mean ± standard deviation) (*p* < 0.05).

Water Loss	HG	CG	*p*-Value
Squeezing loss	0.13 ± 0.10	0.09 ± 0.05	0.35
Dripping loss	0.06 ± 0.03	0.04 ± 0.02	0.42
Cooking loss	0.36 ± 0.32	0.35 ± 0.03	0.40

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
