# Peer review of "Effects of Long-Term Gentle Handling on Behavioral Responses, Production Performance, and Meat Quality of Pigs"

_animals, 2020, doi:10.3390/ani10020330_

Round 1

Reviewer 1 Report

The manuscript has improved notably from the previous one. Now, it offers  a detailed description of the study and of the effects of handling on growth and performance of piglets. Thus, I would recommend acceptance of the paper.

Reviewer 2 Report

The authors have revised the manuscript appropriately and now this can be accepted for publication.

This manuscript is a resubmission of an earlier submission. The following is a list of the peer review reports and author responses from that submission.

Round 1

Reviewer 1 Report

The work presented in the manuscript "Effects of long-term gentle handling on behavioral responses, production performance, and meat quality of pigs", although of interest, is flawed by a deficient design (there is no previous information on the adequacy of the management  tools used and its significance in the data considered), a lack of objective measurements on the stress level of the animals and a lack of significant results. Hence, it is not possible to discern whether the results obtained, mainly linked to a lack of significant effects, are biased by the design and the proper study or real.

Reviewer 2 Report

In general, I am happy with this manuscript. The authors have presented a detailed account of an interesting topic and the manuscript after appropriate small-scale revision can be accepted.

Specific comments are below.

Introduction can be abridged.

Details of the assessment scores and parametres for both avoidance and approach should be provided in a table.

For sections 3.2 and 3.3 only brief comments should be given in the text and detailed results should be provided in relevant tables.

A new section 4.4 should be introduced to discuss the findings in view of the relevant EU and USA legislation.

Finally, English language MUST be improved by any means.